# Phenotypic Characterization and Prevalence of Carbapenemase-Producing *Pseudomonas aeruginosa* Isolates in Six Health Facilities in Cameroon

Cecile Ingrid Djuikoue [1,*], Paule Dana Djouela Djoulako [2], Hélène Valérie Same Njanjo [3], Christiane Possi Kiyang [1], Feline Leina Djantou Biankeu [3], Celianthe Guegang [3], Andrea Sheisley Didi Tchouotou [1], Rodrigue Kamga Wouambo [4], Benjamin D. Thumamo Pokam [5], Teke Apalata [6] and Katy Jeannot [7,8]

1 Department of Microbiology, Faculty of Health Sciences, Université des Montagnes, Bangangté BP 208, Cameroon
2 Faculty of Medicine, Sorbonne Université, 75013 Paris, France
3 Estuary Academic and Strategic Institute (IUEs/INSAM), Douala 4100, Cameroon
4 Division of Hepatology, Department of Medicine II, Leipzig University Medical Center, University of Leipzig, 04103 Leipzig, Germany
5 Department of Medical Laboratory Science, Faculty of Health Sciences, University of Buea, Buéa P.O. Box 63, Cameroon
6 Faculty of Health Sciences & National Health Laboratory Services, Walter Sisulu University, Mthatha 5117, South Africa
7 UMR6249 Chrono-Environnement, Faculté de Médecine-Pharmacie, Université de Bourgogne-Franche Comté, 25030 Besançon, France
8 Centre National de Référence de la Résistance aux Antibiotiques, Centre Hospitalier Universitaire de Besançon, 3 Bd Fleming, 25030 Besançon, France
* Correspondence: djuikoe1983@yahoo.fr

**Abstract:** *Pseudomonas aeruginosa* is a Gram-negative opportunistic pathogen with a great ability to adapt to stress, in particular, to the selective pressure of antibiotics in the hospital environment. This pathogen constitutes a real public health concern, especially in low- and middle-income countries. In Cameroon, little is known about the drug resistance patterns of *Pseudomonas aeruginosa*. This study sought to determine the prevalence of *Pseudomonas aeruginosa* strains producing carbapenemases in six health facilities in the center, littoral, and west regions of Cameroon. An analytical cross-sectional study was conducted over a four-month period from July to October 2021. All *Pseudomonas aeruginosa* or suspected strains isolated from pathological products at the bacteriology laboratory of different health facilities were systematically collected and underwent a re-identification. After growing on cetrimide agar and successfully subculturing on nutrient agar, an oxidase test was performed on pure colonies, followed by biochemical identification (API 20NE system) of the bacterial suspension (0.5McFarland standard). Drug susceptibility testing for the detection of extended-spectrum beta-lactamases of overproduced inducible cephalosporinases and carbapenemases was performed according to adequate standard procedures. Of the 468 isolates collected, 347 (74.14%) were confirmed *Pseudomonas aeruginosa* after re-identification, of which 34.49% (120/347) produced inducible cephalosporinases (CAZ^R and C/T^S) and 32.26% (112/347) extended-spectrum beta-lactamases. The prevalence of carbapenemase-producing *P. aeruginosa* (IMP^R and C/T^R) was 25.07% (87/347), with 17.24% (15/87) class A and 82.76% (72/87) class B. A high rate of resistance to penicillin (piperacillin: 70.58% and ticarcillin: 60.24%) was observed. We also noted a 34.49% resistance to ceftazidime, 30.22% to imipenem against 37.02% to meropenem, and 25.1% to ceftolozane/tazobactam (C/T). These strains also exhibited 79.57% resistance to quinolones and about 26% to aminoglycoside families. Multivariate analysis revealed that carbapenemase-producing *Pseudomonas aeruginosa*-related infections were significantly associated with hospitalization ($p = 0.04$), maternity ($p = 0.03$), surgery ($p = 0.04$), and intensive care wards ($p = 0.04$). This study highlighted a high burden of resistant strains of carbapenemase-producing *Pseudomonas aeruginosa*. Surveillance should be intensified to prevent the dissemination and spread of these strains.

**Keywords:** *Pseudomonas aeruginosa*; antibiotic resistance; carbapenemases; hospitals; Cameroon

## 1. Introduction

*Pseudomonas aeruginosa* (PA) is a Gram-negative bacillus (GNB), non-Enterobacteriaceae, and ubiquitous organism mostly encountered in the hospital environment. It is an opportunistic bacterium with little or no virulence in healthy humans [1]. However, the development of PA infections is usually related to patient weakness, hospital acquisition, and the use of indwelling devices, as well as other invasive procedures [2]. These factors are particularly relevant among immunocompromised patients due to their underlying conditions and frequent contact with the healthcare system [3]. For instance, high mortality rates have been reported for PA infections in cancer patients, particularly when multidrug-resistant (MDR) strains are involved [4]. On the other hand, the common prescription of broad-spectrum antibiotics during episodes of febrile neutropenia and other indications of empirical therapy, prolonged antibiotic prophylaxis, and invasive procedures are well-established factors contributing to the colonization by the multidrug-resistant Gram-negative bacilli (MDR-GNB). The higher prevalence of MDR-GNB infection in this patient population, in turn, worsens the prognosis and makes the treatment difficult, which often implies the use of second-line agents with higher toxicity (e.g., polymyxins) [5].

Resistance to beta-lactams occurs through the production of restricted or extended-spectrum beta-lactamases (ESBL), including TEM (Temoneira) type ESBL, SHV (Sulfhydryl-Variable), CTX-M (Cefotaximase-Munich), PER (*Pseudomonas* Extended Resistance), VEB (Vietnam Extended-spectrum Beta-lactamase), GES (Guyana Extended-spectrum Beta-lactamase), and OXA (Oxacillinase) [6]. A less common but increasing mechanism of carbapenem resistance among PA isolates currently is the production of carbapenemases [7]. This mechanism of carbapenem resistance is important because it significantly alters the efficacy of commonly used antipseudomonal agents, including ceftazidime, cefepime, piperacillin–tazobactam, as well as the newly introduced beta-lactam/beta-lactamase inhibitor combinations such as ceftolozane–tazobactam, imipenem–relebactam, and ceftazidime–avibactam [8].

*P. aeruginosa* isolates have been reported to contain a wide variety of carbapenemases globally. These include essentially KPC (*Klebsiella pneumoniae* Carbapenemase), GES (Guiana Extended-Spectrum), IMP (active-on-imipenem), VIM, NDM (New Delhi Metallo-beta-lactamase), and SPM in Latin America [9]; IMP, and NDM in the Arabian Peninsula; [8] and KPC, NDM, VIM, and IMP in the United States [10]. The diversity and emerging prevalence of carbapenemases producers among carbapenem-resistant strains of *P.aeruginosa* (CR-PA) have been recently highlighted in the multi-national ERACE-PA Surveillance Program [11]. Of the 807 CR-PA collected over 2019–2021 from 17 health facilities in 12 countries, 33% were tested phenotypically carbapenemase-positive (using the mCIM method) and 86% of these were genotypically positive (with VIM and GES being most prevalent) [12]. CR-PA pathogens are among the leading cause of hospital epidemics throughout the world, and nosocomial infections in intensive care units (ICU), such as pneumonia, surgical infections, urinary tract infections, and bacteremia and thus, represent a significant threat to public health [13–15].

A large international observational study of infections in the ICU found a prevalence of 16.2% *P. aeruginosa* in-patient infections and 23% in all ICU-acquired infections, among which respiratory-associated *P. aeruginosa* infections were the most common [16]. In fact, it accounts for 10% to 20% of isolates in ventilator-associated pneumonia (VAP) [17] and is also the common cause of nosocomial urinary tract infections (UTIs), particularly catheter-associated urinary tract infections (CAUTIs) [18].

In Africa, the incidence of carbapenemase-producing *Pseudomonas aeruginosa* in nosocomial infections has increased over the past 40 years [19]. In 2020, 21.36% of nosocomial infections were attributed to carbapenemase-producing *Pseudomonas aeruginosa* [20], including 5.1% in the Congo-Brazzaville and 9.1% in the Democratic Republic of Congo [21].

However, in Cameroon, few studies have been carried out across the country on the production of carbapenemase-producing strains of *Pseudomonas aeruginosa*; hence, the objective of this study is to determine the frequency of carbapenemase-producing strains of *Pseudomonas aeruginosa* in six health facilities in Cameroon.

## 2. Materials and Methods

### 2.1. Type, Site, and Duration of Study

A cross-sectional and analytical study was carried out during a four-month period from 7 July to 28 October 2021. Isolates were collected from six hospitals located in the three most crowded regions of Cameroon: three health facilities from the center region (the Military Hospital of Yaoundé, the Saint Martin de Porres Dominican Hospital Center, and the Referral Teaching Hospital of Yaoundé), one from the littoral region (Laquintinie Hospital), and lastly two from the west region (Ad-Lucem Hospital of Banka and the District Hospital of Dschang). The strains re-identification and downstream analyses were carried out at the laboratory setting of Saint Martin de Porres Dominican Hospital Center.

### 2.2. Sampling Method and Selection Criteria

During the study period, all PA or suspected PA strains isolated from pathological specimens (pus, wounds, probe tip, urine, blood, effusion fluid, and endocervical swab) at each bacteriology laboratory of our study sites were systematically collected, stored, and later included in the study for the upcoming re-identification and analyses. Non-confirmed cases of *P. aeruginosa* after re-identification and strains with a lack of useful clinical information were excluded from the study.

### 2.3. Re-Identification and Samples Processing

Subculture of collected isolates: The isolates were inoculated around the flame on cetrimide agar, then followed by nutrient agar by the streaking method. Identification: The colonies obtained were firstly subjected to macroscopic examination (description of the size, color, and appearance of the colonies), followed by an oxidase test (oxidase+) on one pure colony and a biochemical identification using the miniaturized API 20NE system. The latter was performed on a bacterial suspension (using the 0.5McFarland standard). Drug Susceptibility Testing (DST): Subsequently, drug susceptibility testing by the diffusion method on Mueller–Hinton was carried out according to the Antibiogram Committee of the French Society of Microbiology (CA-SFM-2021v.1.0). About 17 discs of antibiotics from different families (BIO-RAD, Marnes-la-coquette, France) were tested using the following concentrations: β-lactams (piperacillin—30 μg, piperacillin + tazobactam—30-6 μg, ticarcillin—75 μg, ticarcillin + clavulanic acid—75-10 μg, ceftolozan + tazobactam—30-10 μg, cefotaxime—30 μg, ceftazidime—10 μg, cefepime—30 μg, aztreonam—30 μg, imipenem—10 μg, and meropenem—10 μg), aminoglycosides (gentamicin—10 μg, amikacin—30 μg, and tobramycin—10 μg), fluoroquinolones (ciprofloxacin—5 μg and levofloxacin—5 μg), and polymyxin (colistin). A pure bacterial suspension was seeded on Mueller–Hinton agar by the swab method, and antibiotic discs were then placed at a distance of 25 to 30 mm from each other. The seeded plates were then incubated at 37 °C for 24 h. Detection of Extended Spectrum β-lactamase (ESBL): This was performed according to the CLSI guidelines [22]. All of the 347 *Pseudomonas aeruginosa* isolates were tested for ceftazidime, aztreonam, and cefepime disks with and without clavulanic acid. A difference of ≥5 mm between the inhibition diameter zone of either the cephalosporin disks or their respective cephalosporin–clavulanate disk was taken to be the phenotypic confirmation of ESBL production [23]. Ceftazidime (100 μg), cefepime (30 μg), aztreonam (30 μg), and ceftazidime/clavulanic acid (10/4 μg) disks were used. Detection of AmpC overproduction: After the DST, presumptive detection of AmpC β-lactamases was also performed on 347 confirmed *Pseudomonas aeruginosa* isolates using ceftazidime and ceftolozan–tazobactam disks as described by the National Reference Center of antibiotic resistance (CNR, CHU Besançon, France). All isolates of *P. aeruginosa* resistant to ceftazidime and sensitive to ceftolozan–tazobactam were reported "positive for AmpC beta-lactamases". De-

tection/Classification of carbapenemases: A decrease in sensitivity to imipenem (d < 20 mm) and a resistance to the ceftolozan–tazobactam association (d < 24 mm) with ceftolozan 4 mg/L was suggestive of carbapenemase production (class A/class B) according to CA-SFM-2021v.1.0. In addition, an EDTA inhibition test (if positive) was then used to confirm the production of metallo beta-lactamase (carbapenemase class B).

### 2.4. Data Evaluation and Analysis

The different variables and results obtained were recorded in Excel 2013 software and then analyzed with the statistical software, Statview 5.0 (SAS Institute Inc., Cary, NC, USA). The analysis included the calculation of the frequency and their intervals at 95% (for qualitative variables) and the mean or the median (for quantitative variables). Odds ratios were used to determine the risk factors associated with *Pseudomonas aeruginosa*-producing carbapenemases. A chi-square test was used to compare the proportion of categorical variables, and a value of $p < 0.05$ was considered statistically significant.

### 2.5. Ethical Considerations

An ethical clearance was issued by the institutional research ethics committee of the Université des Montagnes (Authorization N°2021/171/UDM/PR/CIE; 15 October 2021), and research authorizations from the directors of each hospital were obtained.

### 3. Results

### 3.1. Description of the Source Characteristics of the Isolates Collected

From the 468 strains collected at the baseline, 347 (74.14%) were finally confirmed as *Pseudomonas aeruginosa* after re-identification (Table 1).

**Table 1.** General characteristics of *P. aeruginosa* isolates.

| Characteristics | Effective (n) | Percentage (%) |
|---|---|---|
| Collection Sites | | |
| Center region | 89 | 25.64 |
| Littoral region | 102 | 29.40 |
| West region | 156 | 44.96 |
| Age (in years) | | |
| Mean ± Standard Deviation | 34.38 ± 20.67 | |
| Minimum; Maximum | 2 days; 91 years | |
| Sex | | |
| Male | 187 | 53.89 |
| Female | 160 | 46.11 |
| Sex ratio | 1.2 | |
| Healthcare unit | | |
| Surgery | 143 | 41.21 |
| Intensive care | 78 | 22.48 |
| Medicine | 56 | 16.14 |
| Pediatrics | 36 | 10.37 |
| Outpatient | 19 | 5.48 |
| Maternity | 15 | 4.32 |
| Hospitalization | | |
| Yes | 243 | 70.03 |
| No | 104 | 29.97 |
| Samples type | | |
| Pus | 142 | 40.92 |
| Wounds | 113 | 32.56 |
| Probe tip | 45 | 12.96 |
| Urine | 26 | 7.49 |
| Blood | 16 | 4.61 |
| Effusion fluid | 4 | 1.15 |
| Endocervical swab | 1 | 0.31 |
| Treatment | | |
| Yes | 219 | 59.65 |
| No | 128 | 36.88 |

The majority of isolates came from urban areas in the center and littoral regions of Cameroon, 55.04% (191/347), from male patients, 53.89% (187/347). About ¾ of the study participants were hospitalized, 70.03% (243/347), and 63.12% (219/347) were on antibiotic treatment. The surgical and intensive care units were the most represented, with 41.21% (143/347) and 22.48% (78/347), respectively. The samples' type including pus, 40.92% (142/347), and wounds, 32.56% (113/347), were predominant.

### 3.2. Susceptibility of Pseudomonas aeruginosa Isolates to Beta-Lactams

The drug susceptibility profile of *Pseudomonas aeruginosa* to beta-lactam is presented below (Figure 1).

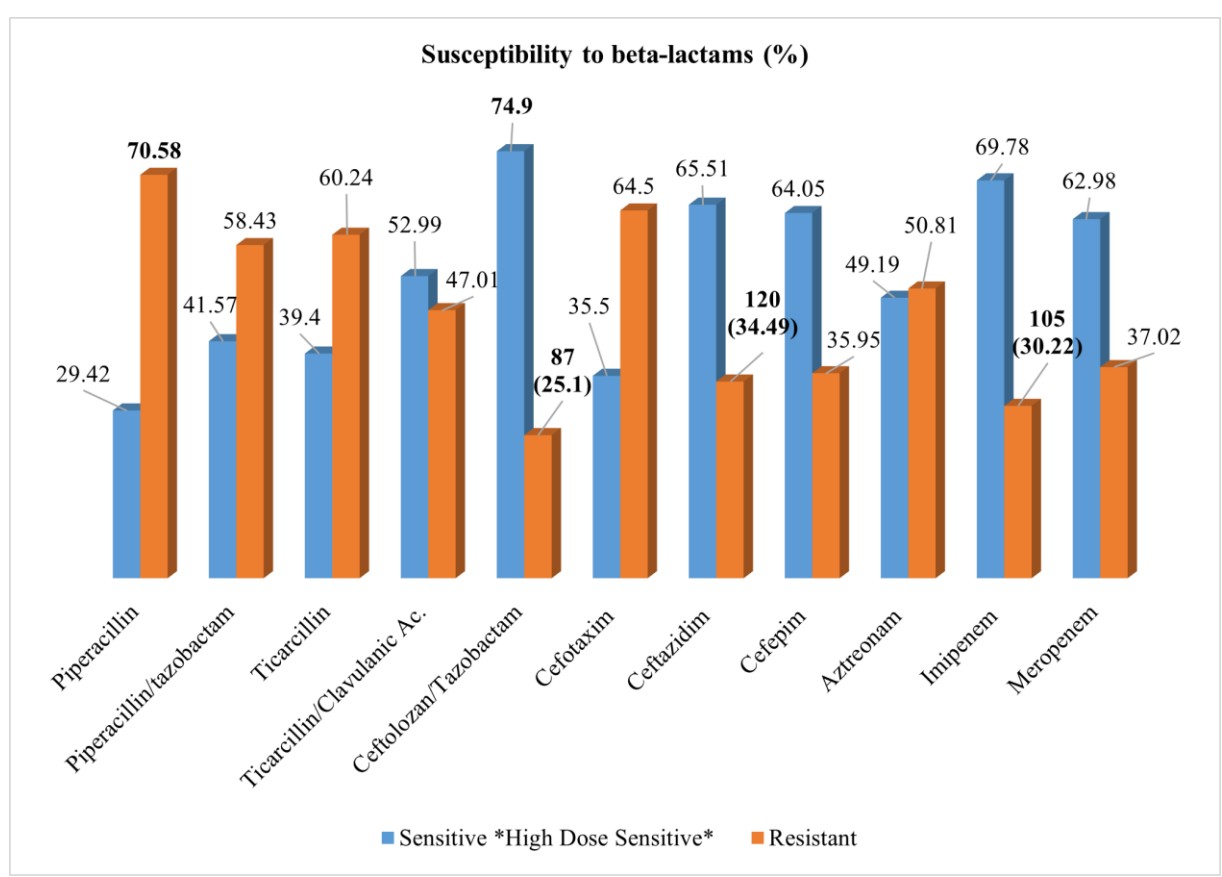

**Figure 1.** Susceptibility profile of *Pseudomonas aeruginosa* isolates to beta-lactam.

The sensitivity profile of *Pseudomonas aeruginosa* showed a high resistance rate to penicillins (piperacillin: 70.58% and ticarcillin: 60.24%) and to piperacillin/tazobactam (58.43%). It was also noted a 34.49% resistance to ceftazidime, 30.22% to imipenem versus 37.02% to meropenem, and lastly, 25.1% to ceftolozan/tazobactam. However, there was a high sensitivity to the ceftolozan/tazobactam (74.90%) and (69.78%) to imipenem (Figure 1).

### 3.3. Resistance Profile of Pseudomonas aeruginosa to Other Families of Antibiotics

Antibiotics belonging to the family of polymyxins, quinolones, and aminoglycosides were tested in order to determine their resistance profile (Figure 2).

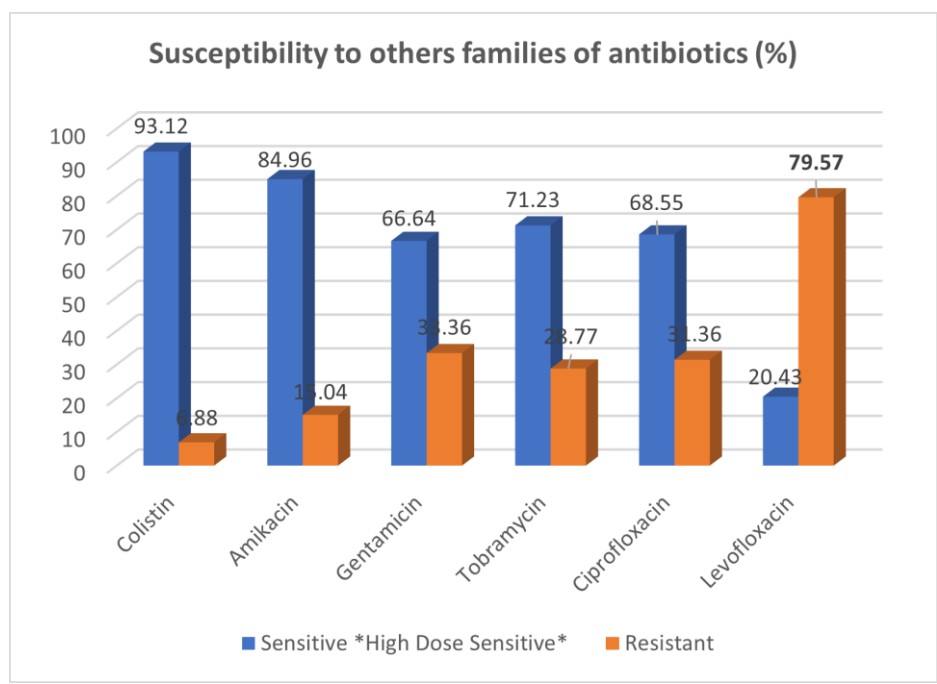

**Figure 2.** Resistance to other families of antibiotics.

There was high resistance to quinolones with 79.57% to levofloxacin but low and very low resistance to aminoglycosides and polymyxins (26% on average and 6.9%, respectively). The majority of the strains were sensitive to colistin (93.12%).

*3.4. Distribution of Pseudomonas aeruginosa Isolates According to Enzyme Production and Carbapenemase Classification*

The production of ESBL, AmpC, and carbapenemase is shown in Figure 3, and the classification of carbapenemases is shown in Figure 4.

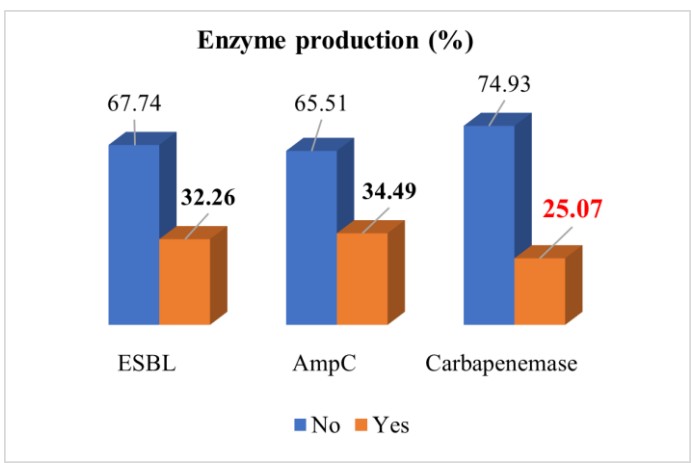

**Figure 3.** Distribution of *Pseudomonas aeruginosa* according to the type of enzyme production.

In this study, up to 32.26% (112/347), PA produced ESBL and 34.49% (120/347) AmpC. The overall prevalence of *Pseudomonas aeruginosa*-producing carbapenemase was 25.07% (87/347) with 82.76% (72/87) of class B (metallo-beta-lactamase) and 17.24% (15/87) of class A.

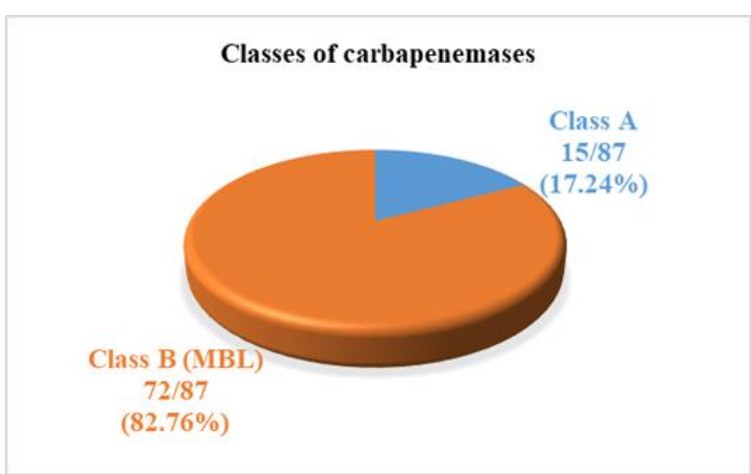

**Figure 4.** Classification of carbapenemases.

*3.5. Co-Resistance Profile of Pseudomonas aeruginosa Producing Carbapenemase to Various Families of Antibiotics*

There was high resistance to quinolones (80.57% to levofloxacin and 48.36% to ciprofloxacin), to aminoglycosides (52.77% to tobramycin, 59.36% to gentamicin, and 44.04% to amikacin) and low resistance to polymyxins (6.9% to colistin). Furthermore, the majority of the isolates were sensitive to colistin (93.12%) (Figure 5).

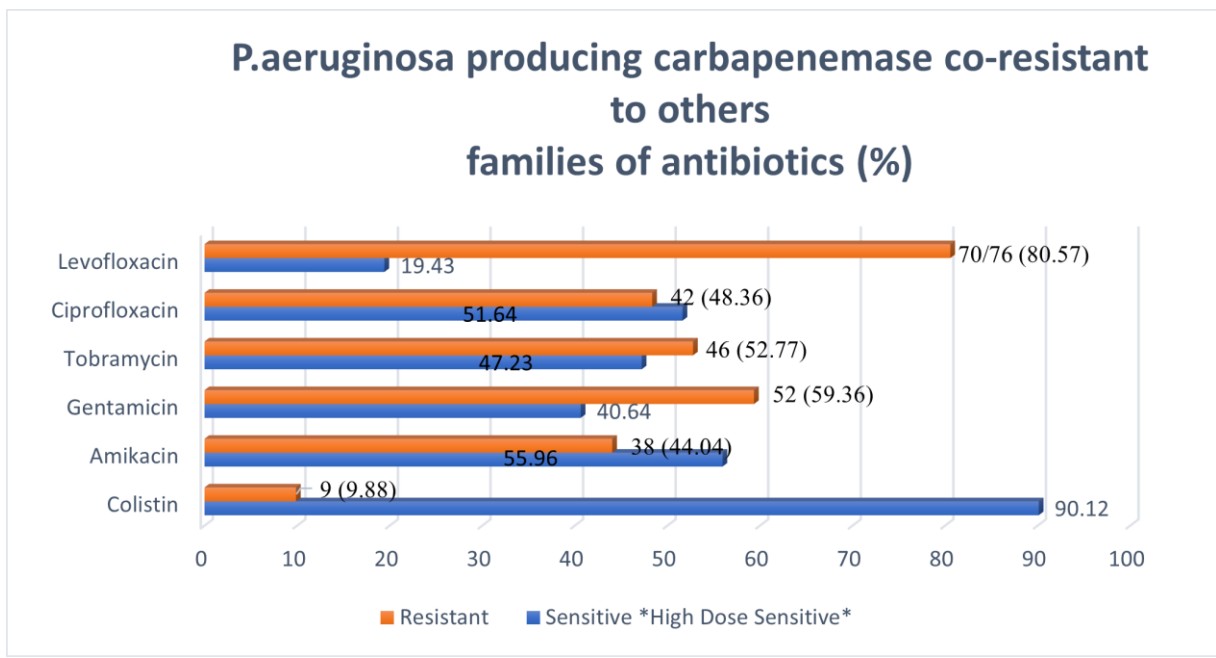

**Figure 5.** Co-resistance of *Pseudomonas aeruginosa* producing carbapenemase to various families of antibiotics.

*3.6. Association between Risk Factors and Carbapenemase-Producing Pseudomonas aeruginosa*

The possible associations between the carbapenemase-producing *Pseudomonas aeruginosa* isolates and source patients' characteristics were evaluated, as shown in Tables 2 and 3.

In univariate analysis, carbapenemase-producing *Pseudomonas aeruginosa* infection was significantly associated with the region (with a higher risk in urban areas) and the surgical and intensive care units (Table 2).

**Table 2.** Univariate analysis of the distribution of carbapenemase-producing *Pseudomonas aeruginosa* infections by the source characteristics of the collected strains.

| Variables | Total (n) | Carbapenemase Productionn (%) | OR | 95% CI | *p*-Value |
|---|---|---|---|---|---|
| **Sampling sites** | | | | | |
| Rural (West region) | 156 | 16 (10.44%) | Ref | | |
| Urban (Center and Littoral region) | 191 | 54 (28.48%) | 1.31 | 0.66–2.59 | 0.0467 |
| **Gender** | | | | | |
| Female | 160 | 30 (18.84%) | Ref | | |
| Male | 187 | 41 (21.74%) | 1.20 | 0.68–2.11 | 0.5355 |
| **Hospitalization** | | | | | |
| No | 104 | 15 (14.44%) | Ref | | |
| Yes | 243 | 56 (22.96%) | 0.94 | 0.51–1.73 | 0.1114 |
| **Treatment** | | | | | |
| No | 128 | 27 (20.91%) | Ref | | |
| Yes | 219 | 46 (20.11%) | 0.95 | 0.53–1.70 | 0.2480 |
| **Healthcare unit** | | | | | |
| External | 19 | 1 (33.56%) | Ref | | |
| Maternity | 15 | 4 (28.57%) | 1.84 | 0.89–4.45 | 0.5785 |
| Medicine | 56 | 11 (18.75%) | 0.75 | 0.32–1.73 | 0.4960 |
| Pediatrics | 36 | 3 (9.68%) | 0.35 | 0.10–1.23 | 0.1003 |
| Surgery | 143 | 48 (6.25%) | 1.88 | 0.99–2.41 | 0.0432 |
| Intensive care | 78 | 17 (22.39%) | 1.4 | 0.04–1.90 | 0.0457 |
| **Sample type** | | | | | |
| probe tip | 45 | 12 (25.64%) | Ref | | |
| Blood | 16 | - | $7.24 \times 10^{-7}$ | - | 0.9789 |
| ascites fluid | 4 | 1 (25%) | 0.97 | 0.19–2.57 | 0.9935 |
| PCV | 1 | - | $7.24 \times 10^{-7}$ | - | 0.9944 |
| Wounds | 113 | 20 (17.53%) | 0.62 | 0.25–1.50 | 0.2859 |
| Pus | 142 | 34 (23.77%) | 0.90 | 0.39–2.08 | 0.8124 |
| Urine | 26 | 5 (18.18%) | 0.64 | 0.18–2.37 | 0.5078 |

**Table 3.** Multivariate analysis of risk factors associated with carbapenemase-producing *Pseudomonas aeruginosa* infection.

| Variables | Adjusted OR | 95% CI | *p*-Value |
|---|---|---|---|
| **Sampling sites** | | | |
| Rural (West) | Ref | | |
| Urban (Center and Littoral) | 1.39 | 0.52–3.68 | 0.5116 |
| **Hospitalization** | | | |
| No | Ref | | |
| Yes | 1.45 | 0.03–1.73 | 0.0456 |
| **Treatment** | | | |
| No | Ref | | |
| Yes | 0.86 | 0.47–1.60 | 0.6435 |
| **Ward** | | | |
| External | Ref | | |
| Medicine | 1.30 | 0.38–4.45 | 0.6795 |
| Maternity | 1.75 | 0.02–1.73 | 0.0365 |
| Pediatrics | 0.35 | 0.10–1.23 | 0.1003 |
| Surgery | 1.41 | 0.32–1.71 | 0.0434 |
| Intensive care | 1.29 | 0.05–1.45 | 0.0484 |

In the multivariate analysis using logistic regression, a significant association of carbapenemase-producing *Pseudomonas aeruginosa* infection to patient hospitalization, maternity, intensive care, and surgical units was found (Table 3).

## 4. Discussion

In order to fight against the emergence and spread of multiresistant bacteria, our study aimed to determine the frequency of carbapenemase-producing *Pseudomonas aeruginosa* from six health facilities. The study took place in three regions of Cameroon (central, littoral, and western regions). Out of 468 collected strains, 74.16% were confirmed *Pseudomonas aeruginosa* isolates. The majority of the isolates came from urban areas (55.04% urban vs. 44.96% rural zone). These results are similar to those of Zahraa et al. in 2019 in Iraq, who reported a prevalence of *Pseudomonas aeruginosa* isolates equal to 55.38% in urban areas and 44.62% in rural areas [24]. This could be explained by the problem of poor sanitation and overcrowding of hospitals in some urban cities. Moreover, up to 70.03% of the isolates were from hospitalized patients. *Pseudomonas aeruginosa* has been shown to colonize the hospital environment and to be resistant to many antibiotics, making its elimination difficult. Furthermore, about 41% of the samples in our study were pus. Gonsu et al., in their study carried out in the city of Yaoundé in 2015, also found that *Pseudomonas aeruginosa* were more isolated in hospitalized patients, with 17.6% from pus samples [25].

From drug susceptibility testing, the *Pseudomonas aeruginosa* had a high resistance to beta-lactams, especially to the penicillins, with 70.48% to piperacillin, 60.24% to ticarcillin, and 58.43% to piperacillin/tazobactam. Indeed, penicillins and third-generation cephalosporins are currently increasingly used in humans and animals and are easily accessible in street pharmacies at very low costs [26]. One of the most striking features of *P. aeruginosa* is its outstanding capacity for developing acquired antimicrobial resistance to nearly all available antipseudomonal agents through the selection of chromosomal mutations [27,28]. The other resistance mechanisms can be classified into intrinsic and adaptive resistance [29]. The intrinsic resistance of *P. aeruginosa* includes low outer membrane permeability, expression of efflux pumps that expel antibiotics out of the cell, and the production of antibiotic-inactivating enzymes [30]. The adaptive involves the formation of biofilm in the lungs of infected patients where the biofilm serves as a diffusion barrier to limit antibiotic access to the bacterial cells [31]. In addition, multidrug-tolerant persister cells that are able to survive an antibiotic attack can form in the biofilm; these cells are responsible for prolonged and recurrent infections in CF patients [32]. Furthermore, about 16.39% of PA were resistant to 34.49% ceftazidime, 30.22% imipenem, 37.02% meropenem, and 25.1% ceftolozan/tazobactam. These high resistance rates pose a great challenge given that they belong to the last classes of antibiotics and are therefore used as the last therapeutic alternatives in cases of antibiotic therapy failure.

The antibiogram results also showed co-resistances to the family of aminoglycosides (15.04% to amikacin, 33.36% to gentamicin, and 24.75% to tobramycin), quinolones (77.59% to levofloxacin and 29.1% to ciprofloxacin), and polymyxin (3.6% to colistin). Similar trends were already reported by Moctar et al. in Cameroon in 2019 [33]. Several studies suggested that mutations play very crucial roles in developing drug resistance and cross-resistance through the selection of chromosomal mutations [27,28]. In fact, the accumulation of several chromosomal mutations leads to the emergence of multidrug-resistant (MDR), extensively drug-resistant (XDR), or even pan-antibiotic-resistant (PDR) strains, which can be responsible for notable epidemics in the hospital setting [34,35].

According to different production phenotypes of beta-lactamases, the overproduction of the inducible plasmid cephalosporinases was the most represented at 34.49% and the ESBL production at 32.26%. These results are similar to those of Massri et al. in 2016 who showed that the most represented resistant phenotype in *Pseudomonas aeruginosa* was the production of the AmpC inducible plasmid cephalosporinase [19]. Equally, Ankur et al., in their study, had a 26.6% prevalence of ESBL-producing *Pseudomonas aeruginosa* [26]. In fact, *Pseudomonas aeruginosa* naturally produces a cephalosporinase AmpC and mutation-

dependent overproduction of intrinsic β-lactamase AmpC is considered the main cause of resistance of clinical strains to antipseudomonal penicillins and cephalosporins [36].

Of the 347 isolates of *Pseudomonas aeruginosa*, the carbapenemase producers represented 25.07%, of which 17.24% were class A carbapenemase and 82.76% were class B. These results are higher compared to the study of Castanhiera et al. in 2014 who found 20% of *Pseudomonas aeruginosa* producing carbapenemases [37] and that of Alkudhairy et al. in 2020 who found 10.3% of *Pseudomonas aeruginosa* producing class B carbapenemases [38]. These findings buttress the evolution of carbapenemase-producing strains of *P. aeruginosa* species globally.

Populations in this study were more at risk of contracting a carbapenemase-producing *Pseudomonas aeruginosa* infection in urban cities of the center and littoral region than in rural areas of the west region of Cameroon. This result is similar to that of Stije et al. in 2014 in Sub-Saharan Africa, who noted a greater rate of multidrug-resistant bacteria in urban areas (73%) compared to rural ones [39]. Although self-medication is common in low- and middle-income countries, the high population density, the overcrowded hospitals, as well as the great number of hospitals favor the development of resistant bacteria and nosocomial infections in urban areas [40]. In the multivariate analysis, carbapenemase-producing *Pseudomonas aeruginosa* infection was significantly associated with hospitalized patients in the maternity ward and surgical and intensive care units. Indeed, it has been shown that linen dressing replacements in maternity wards are not always conducted daily, which triggers the colonization of wounds (such as cesarean sections), as well as long-term urinary catheter postpartum bearers, by likely drug-resistant opportunistic pathogens such as *Pseudomonas aeruginosa* and thus, promoting recurrent infections within the ward [25].

## 5. Limitations

The limitation of this study was the inability to carry out a molecular characterization of the *Pseudomonas aeruginosa* isolates due to the lack of funding.

## 6. Conclusions

This study highlighted a high prevalence of *Pseudomonas aeruginosa*-producing carbapenemase, ESBL, and AmpC. The carbapenemase-producing *Pseudomonas aeruginosa* infections were more frequent in urban areas, hospitalized patients from the maternity ward, and surgical and intensive care units. However, colistin and ceftolozan/tazobactam were antibiotics remaining susceptible to *P. aeruginosa*-resistant strains to B-lactams. Continuous drug-resistant monitoring and preventive measures could help to prevent and curb the dissemination of PA resistance genes, especially in hospital settings.

**Author Contributions:** C.I.D. conceived the project and designed the study. P.D.D.D. searched relevant literature, scrutinized all relevant information, and drafted the manuscript. C.I.D., P.D.D.D. and B.D.T.P. conducted and coordinated the field study. H.V.S.N., C.P.K., F.L.D.B., C.G. and A.S.D.T. collected and processed the samples and data. C.I.D., P.D.D.D., B.D.T.P. and RKW analyzed the data. C.I.D., R.K.W., B.D.T.P., T.A. and K.J. revised the manuscript. All authors have read and agreed to the published version of the manuscript.

**Funding:** Self-financing.

**Institutional Review Board Statement:** The study was approved by the Institutional Human Health Research ethics committee of the Université des Montagnes (*Authorization N° 2021/171/UDM/PR/CIE*; 15 October 2021). We obtained research authorizations issued by the directors of the various hospitals (*Authorization N° Ref 224/21/FALC/HBB/MC/CSAF/SG/LD*; 4 August 2021); (*Authorization N° 210211/DV/M IMDEF/DSM/RSM1/HMR1/12*; 6 October 2021); (*Authorization N° 202/AR/MSP/DRSPO/DSD/HDD*; 13 August 2021).

**Informed Consent Statement:** Not applicable.

**Data Availability Statement:** All data generated or analysed in the course of this study are included in this manuscript.

**Acknowledgments:** The authors are grateful to the directors and staff of the various hospitals.

**Conflicts of Interest:** The authors declare no conflict of interest.

**Consent for Publication:** All authors consented to the publication.

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
