# Peer review of "Phenotypic Characterization and Prevalence of Carbapenemase-Producing Pseudomonas aeruginosa Isolates in Six Health Facilities in Cameroon"

_2673-8430, doi:10.3390/biomed3010006_

Round 1

Reviewer 1 Report

I have five major comments.

Major comments:

1.      Authors examined that the production of ESBL and AmpC was examined by antibiotic susceptibility test. If authors would like to mention enzyme production, authors should measure the enzyme production by direct method such as ELISA. Probably, in this part, authors should rewrite.

2.      How did authors measure resistance to antimicrobial agants ?

3.      It would be more interesting that genotypic analysis is added for ESBL.

4.      Statistical method should be described in the manuscript.

5.      The format of manuscript is not consistent. Authors carefully revised the manuscript and have the manuscript proofread by a professional English company recommend by MDPI.

Author Response

Yes

Can be improved

Must be improved

Not applicable

Does the introduction provide sufficient background and include all relevant references?

(x)

( )

( )

( )

Are all the cited references relevant to the research?

(x)

( )

( )

( )

Is the research design appropriate?

( )

(x)

( )

( )

Are the methods adequately described?

( )

(x)

( )

( )

Are the results clearly presented?

( )

(x)

( )

( )

Are the conclusions supported by the results?

( )

(x)

( )

( )

Comments and Suggestions for Authors

I have five major comments.

Major comments:

  1. Authors examined that the production of ESBL and AmpC was examined by antibiotic susceptibility test. If authors would like to mention enzyme production, authors should measure the enzyme production by direct method such as ELISA. Probably, in this part, authors should rewrite.

Authors: Thanks dear reviewer, we rewrote that part

  1. How did authors measure resistance to antimicrobial agants ?

Authors: Thank you dear reviewer. We added that in the method.

  1. It would be more interesting that genotypic analysis is added for ESBL.

Authors: Dear reviewer, the genotypic analysis has been added as the major limitation of this study. However all PA isolates have been conserved for further analyses in case of available fundings.

  1. Statistical method should be described in the manuscript.

Authors: Thank you dear reviewer. We did it

  1. The format of manuscript is not consistent. Authors carefully revised the manuscript and have the manuscript proofread by a professional English company recommend by MDPI.

Authors: Done as suggested dear reviewer. Thanks once more.

Reviewer 2 Report

The manuscript entitled "Phenotypic Characterization and Prevalence of Carbapenemase-Producing Pseudomonas aeruginosa Isolates in Six Health Facilities in Cameroon" presented a clinical study on characterizing and dominant infection of Carbapenemase-Producing Pseudomonas aeruginosa bacteria isolated from Six Health Facilities in Cameroon. 

Pros: This is an interesting study, and the data are informative about the prevalent infection and the cause of the infection caused by P.aregenosa. 

The relatively large sample collections (for this kind of data) are also a strength. Overall, this is a concise and good manuscript. The introduction is relevant and statics based. Sufficient details about the previous study results are illustrated for readers to follow the present study rationale. Cons: Though the manuscript is informative, however, there are lots of sentences that need to be made more explicit to me and are grammatically incorrect.

Additionally, the authors have made many inappropriate spelling mistakes from the publication's viewpoint. The authors must present accurate and relevant details about the experimental procedures in the material and method sections; thus, they fail to attract readers. Some intensive modifications in many sentences and adding proper experimental details and references could improve it. This being said, it is difficult for me to assess the current work as some analysis issues must be addressed. In conclusion, the manuscript is suitable for publication after the authors have addressed the following comments and questions. 

Comments: 

1. My first comment is regarding your manuscript title, which has a big mistake; please correct the health spelling in your title. 

2. Another concern is regarding the style of writing Pseudomonas aeruginosa. The authors have used different styles (italics and non-italic) in the entire manuscript. Please use the appropriate style according to bacterial taxonomy. 

Abstract Section

2. Comment: In the abstract, please correct the spelling of cepalosporinases, beta-latamases, ceftazidim, and ceftolozan. These are scientific terms, 

3. Authors- we also note 34.49% .....and 25.1% to

ceftolozan/tazobactam (C/T). 

3. Comment:  Please start with the capital letter after the fill-stop in the abstract section. This sentence is grammatically not correct.

4. Authors- The infections with carbapenemase-producing Pseudomonas... strains in hospitals a threat. 

4. Comment:  Again, this sentence is grammatically incorrect; please use are for infections, not is. 

Introduction Section

5. Authors- Pseudomonas aeruginosa is an opportunistic ... life-threatening in weakened subjects.

5. Comment:  What kind of weakened subjects are prone to infection with bacteria? Please specify it in terms of immune-compromised patients. 

6. Authors- Resistance to beta-lactams ..spectrum beta-lactamases (ESBL) [3].

6. Comment: can you please add some examples of these ESBLs?

7. Authors- Carbapenems of beta-lactam subfamily,

....responsible for more than 4000 deaths annually in Europe

[4].

7. Comment: Are these the current statics, as the reference for this study is not current? Please confirm the statics and mention them again.

8. Authors- Infections caused by these resistant .... responsible for hospital epidemics throughout the world [6].

8. Comment: Can the authors provide some examples of nosocomial infections that would be easy to follow for readers? 

Materials and Methods Section

9. Authors- Were excluded from the study those without confirmed ...those isolated from any other sample than those targeted.

9. Comment: Can you please rewrite this sentence and specify what the authors have done in this experiment? This is very tough to understand and why the sentence started with were.

Samples processing Section

Comment: Authors must elaborate more on this section and miss crucial details that need to be addressed, as this section looks incomplete and blemished. I have many concerns regarding this section; please specify them individually. 

10. What standard microbiological procedures were used to detect Extended Spectrum B-Lactamase (ESBL)? Please describe this in detail, as this is the crucial section of the study. The authors should describe it in complete information.

11. Please rewrite B-lactamase to β-lactamase; this is an inconvenient way of denoting this enzyme.

12. What were the third or fourth-generation cephalosporins used in the experiments to detect ESBL?

13. What do the authors mean by suspecious to overproduce AmpC? Please write it correctly to follow the essence of the experiments. 

14. How the authors detect the overproduction of AmpC. Please describe it elaborately and include your results in the main or supplementary section. 

15. Again, there are many spelling mistakes; please correct them.

16. In the Detection/Classification of carbapenemases section, my major concern is how the authors have conducted these experiments; please describe it. Under which conditions have these experiments been performed? Such as what concentrations they used for experiments and how they identified or considered that the test organism produced ESBL. If you performed synergy tests as stated by the authors, please explain what synergism is and how they perform these tests. Are there any other tests for the same?

 17. What do the authors mean by stating that the formation of a champagne cork characteristic of synergy between a B-lactamase inhibitor and a third or fourth-generation cephalosporin? What is champagne cork characteristic? Please add these results in the supplementary section. Add these detail in the material method section correctly.

18. How the authors performed this experiment? More than simply adding a sing line/sentence is needed to explain this crucial test. The authors should explain it briefly and the what was the outcome of this test; the authors have not included this detail in the results section; please add them to the data (supplementary). 

Result Section:

19 Authors: the surgery department was the most represented with 41.21% ( followed.... of pus (40.92%) and wounds (32.56%).

Comment: Please start the sentence with capital letters. 

20 Authors: The sensitivity profile of Pseudomonas aeruginosa isolates to the different... (71.88%) and 69.78% to imipenem (Figure 1).

Comment: Please rewrite this entire paragraph; this makes no sense to me, and again, after a full stop, start with capital letters. Please note that the authors have made these many grammatical mistakes throughout the manuscript; correct them all. 

20 Authors: Up to 112/347 (32.26%) produced ESBL and 120/347 (34.49%) AmpC. 

Comment: What it means? This sentence needs to be more coherent. Please rewrite this sentence so that one can understand its true intention. 

21. Please provide all the details of the results of material methods experiments with figures (data/test results)

Discussion section:

22. Though the authors have described a high drug resistance of P.aregenosa for b-lactams due to selective pressure, they still need to provide the details of any mutational evidence. I.e., several studies suggested that mutations play very crucial roles in developing drug resistance. Can authors add a brief statement about the type of mutational changes driving drug-resistance phenomena? This would strengthen this section well.

23. Another crucial reason for developing drug resistance in bacteria is due to an environmental stimulus, and it is reversible when the stimulus is removed. This is best described in the case of P. aeruginosa. Can the authors provide detailed mechanisms of adaptive resistance using such a phenomenon in P. aeruginosa? Or any detail about the biofilm formation mechanism in P. aeruginosa regarding this context, specifically P. aeruginosa's persistent infection in cystic fibrosis. Please include these crucial points and studies in the discussion section.

24. Authors: Of the 347 isolates of Pseudomonas aeruginosa, the carbarpenemase producers...traditional treatments which certainly contain antibiotics but with weak active ingredients.

Comment: 25. Please correct the spelling of butresses; it's incorrect in this paragraph.

26. Is this the only reason for incrementing the % of these values of carbarpenemase-producing strains? Please argue this better, as there are several critical mechanisms of carbapenem resistance in P. aeruginosa.

Author Response

Open Review

English language and style

( ) English very difficult to understand/incomprehensible
(x) Extensive editing of English language and style required
( ) Moderate English changes required
( ) English language and style are fine/minor spell check required
( ) I don't feel qualified to judge about the English language and style

Yes

Can be improved

Must be improved

Not applicable

Does the introduction provide sufficient background and include all relevant references?

( )

(x)

( )

( )

Are all the cited references relevant to the research?

( )

(x)

( )

( )

Is the research design appropriate?

( )

(x)

( )

( )

Are the methods adequately described?

( )

( )

(x)

( )

Are the results clearly presented?

( )

( )

(x)

( )

Are the conclusions supported by the results?

( )

( )

(x)

( )

Comments and Suggestions for Authors

The manuscript entitled "Phenotypic Characterization and Prevalence of Carbapenemase-Producing Pseudomonas aeruginosa Isolates in Six Health Facilities in Cameroon" presented a clinical study on characterizing and dominant infection of Carbapenemase-Producing Pseudomonas aeruginosa bacteria isolated from Six Health Facilities in Cameroon. 

Pros: This is an interesting study, and the data are informative about the prevalent infection and the cause of the infection caused by P.aregenosa. 

The relatively large sample collections (for this kind of data) are also a strength. Overall, this is a concise and good manuscript. The introduction is relevant and statics based. Sufficient details about the previous study results are illustrated for readers to follow the present study rationale. Cons: Though the manuscript is informative, however, there are lots of sentences that need to be made more explicit to me and are grammatically incorrect.

Additionally, the authors have made many inappropriate spelling mistakes from the publication's viewpoint. The authors must present accurate and relevant details about the experimental procedures in the material and method sections; thus, they fail to attract readers. Some intensive modifications in many sentences and adding proper experimental details and references could improve it. This being said, it is difficult for me to assess the current work as some analysis issues must be addressed. In conclusion, the manuscript is suitable for publication after the authors have addressed the following comments and questions. 

Comments: 

  1. My first comment is regarding your manuscript title, which has a big mistake; please correct the health spelling in your title. 

Authors: Sorry for that typing mistake dear reviewer. We did correct it.

  1. Another concern is regarding the style of writing Pseudomonas aeruginosa. The authors have used different styles (italics and non-italic) in the entire manuscript. Please use the appropriate style according to bacterial taxonomy. 

 Authors: Thank you dear reviewer for that remark. We corrected as suggested.

Abstract Section

  1. Comment: In the abstract, please correct the spelling of cepalosporinases, beta-latamases, ceftazidim, and ceftolozan. These are scientific terms, 

 Authors: Corrected dear reviewer. Thanks

  1. Authors-we also note 34.49% .....and 25.1% to

ceftolozan/tazobactam (C/T). 

  1. Comment:Please start with the capital letter after the fill-stop in the abstract section. This sentence is grammatically not correct.

 Authors: Thanks dear reviewer.

  1. Authors-The infections with carbapenemase-producing Pseudomonas... strains in hospitals a threat. 
  2. Comment:Again, this sentence is grammatically incorrect; please use are for infections, not is. 

Authors: Thank you dear reviewer for this remark. We did it.

Introduction Section

  1. Authors-Pseudomonas aeruginosa is an opportunistic ... life-threatening in weakened subjects.
  2. Comment:What kind of weakened subjects are prone to infection with bacteria? Please specify it in terms of immune-compromised patients. 

Authors: Thank you dear reviewer for this remark. We specified it.

  1. Authors- Resistance to beta-lactams ..spectrum beta-lactamases (ESBL) [3].
  2. Comment:can you please add some examples of these ESBLs?

Authors : Thank you dear reviewer for this remark. We added them as requested.

  1. Authors- Carbapenems of beta-lactam subfamily,

....responsible for more than 4000 deaths annually in Europe

[4].                          

  1. Comment: Are these the current statics, as the reference for this study is not current? Please confirm the statics and mention them again.

Authors: Thank you dear reviewer. We revised it.

  1. Authors- Infections caused by these resistant .... responsible for hospital epidemics throughout the world [6].
  2. Comment: Can the authors provide some examples of nosocomial infections that would be easy to follow for readers? 

 Authors: Thank you dear reviewer for this remark. We added as requested.

Materials and Methods Section

  1. Authors- Were excluded from the study those without confirmed ...those isolated from any other sample than those targeted.
  2. Comment: Can you please rewrite this sentence and specify what the authors have done in this experiment? This is very tough to understand and why the sentence started with were.

Authors: We rewrote that sentence accordingly. Thanks

Samples processing Section

Comment: Authors must elaborate more on this section and miss crucial details that need to be addressed, as this section looks incomplete and blemished. I have many concerns regarding this section; please specify them individually. 

  1. What standard microbiological procedures were used to detect Extended Spectrum B-Lactamase (ESBL)? Please describe this in detail, as this is the crucial section of the study. The authors should describe it in complete information.

Authors: Thank you dear reviewer. We revised it.

  1. Please rewrite B-lactamase to β-lactamase; this is an inconvenient way of denoting this enzyme.

Authors: Thank you dear reviewer for this remark we revised it.

  1. What were the third or fourth-generation cephalosporins used in the experiments to detect ESBL?

Authors: dear reviewer, as you can see in the method, we used Ceftazidime und Cefepime disks to detect ESBL.

  1. What do the authors mean by suspecious to overproduce AmpC? Please write it correctly to follow the essence of the experiments.

Authors:  We revised it as requested dear reviewer

  1. How the authors detect the overproduction of AmpC. Please describe it elaborately and include your results in the main or supplementary section.

Authors: Done dear reviewer.  Thanks

  1. Again, there are many spelling mistakes; please correct them.

Authors: dear reviewer we revised the entire manuscript and corrected them as requested.

  1. In the Detection/Classification of carbapenemases section, my major concern is how the authors have conducted these experiments; please describe it. Under which conditions have these experiments been performed? Such as what concentrations they used for experiments and how they identified or considered that the test organism produced ESBL. If you performed synergy tests as stated by the authors, please explain what synergism is and how they perform these tests. Are there any other tests for the same?

Authors: dear reviewer, we described it in the method as requested.

  1. What do the authors mean by stating that the formation of a champagne cork characteristic of synergy between a B-lactamase inhibitor and a third or fourth-generation cephalosporin? What is champagne cork characteristic? Please add these results in the supplementary section. Add these detail in the material method section correctly.

Authors: dear reviewer we rewrote that section for a better understanding

  1. How the authors performed this experiment? More than simply adding a sing line/sentence is needed to explain this crucial test. The authors should explain it briefly and the what was the outcome of this test; the authors have not included this detail in the results section; please add them to the data (supplementary). 

Authors: dear reviewer, we described it in the method section of the revised version of the manuscript.

Result Section:

19 Authors: the surgery department was the most represented with 41.21% ( followed.... of pus (40.92%) and wounds (32.56%).

Comment: Please start the sentence with capital letters. 

Authors: Thank you dear reviewer for this remark we did it.

20 Authors: The sensitivity profile of Pseudomonas aeruginosa isolates to the different... (71.88%) and 69.78% to imipenem (Figure 1).

Comment: Please rewrite this entire paragraph; this makes no sense to me, and again, after a full stop, start with capital letters. Please note that the authors have made these many grammatical mistakes throughout the manuscript; correct them all. 

20 Authors: Up to 112/347 (32.26%) produced ESBL and 120/347 (34.49%) AmpC. 

Comment: What it means? This sentence needs to be more coherent. Please rewrite this sentence so that one can understand its true intention.

 Authors: dear reviewer we rewrote them as requested

  1. Please provide all the details of the results of material methods experiments with figures (data/test results)

 Authors: done dear reviewer. Thanks

Discussion section:

  1. Though the authors have described a high drug resistance of P.aregenosa for b-lactams due to selective pressure, they still need to provide the details of any mutational evidence. I.e., several studies suggested that mutations play very crucial roles in developing drug resistance. Can authors add a brief statement about the type of mutational changes driving drug-resistance phenomena? This would strengthen this section well.

Authors: Thanks very much for that comment dear reviewer, we added it.

  1. Another crucial reason for developing drug resistance in bacteria is due to an environmental stimulus, and it is reversible when the stimulus is removed. This is best described in the case of P. aeruginosa. Can the authors provide detailed mechanisms of adaptive resistance using such a phenomenon in P. aeruginosa? Or any detail about the biofilm formation mechanism in P. aeruginosa regarding this context, specifically P. aeruginosa's persistent infection in cystic fibrosis. Please include these crucial points and studies in the discussion section.

Authors: done as requested dear reviewer, thanks

  1. Authors: Of the 347 isolates of Pseudomonas aeruginosa, the carbarpenemase producers...traditional treatments which certainly contain antibiotics but with weak active ingredients.

Comment: 25. Please correct the spelling of butresses; it's incorrect in this paragraph.

Authors: Corrected, thanks dear reviewer

  1. Is this the only reason for incrementing the % of these values of carbarpenemase-producing strains? Please argue this better, as there are several critical mechanisms of carbapenem resistance inP. aeruginosa.

Authors: thanks dear reviewer, we did it

Round 2

Reviewer 1 Report

In the text, many typos can be found. For example, table 2 in the text should be Table 2.

Please proofread the manuscript.

Figures: The format of figures (i.e., bar graphs) should be formatted in the same manner. One figure is 3D bar graph, and other is 2D bar graph.

Table 2: The horizontal lines on the top of the table is not normal. Thickness of horizontal lines should be corrected.

Table 3 : Thickness of horizontal lines should be corrected.

Author Response

Open Review

English language and style

( ) English very difficult to understand/incomprehensible
( ) Extensive editing of English language and style required
(x) Moderate English changes required
( ) English language and style are fine/minor spell check required
( ) I don't feel qualified to judge about the English language and style

Yes

Can be improved

Must be improved

Not applicable

Does the introduction provide sufficient background and include all relevant references?

(x)

( )

( )

( )

Are all the cited references relevant to the research?

(x)

( )

( )

( )

Is the research design appropriate?

(x)

( )

( )

( )

Are the methods adequately described?

(x)

( )

( )

( )

Are the results clearly presented?

(x)

( )

( )

( )

Are the conclusions supported by the results?

(x)

( )

( )

( )

Comments and Suggestions for Authors

In the text, many typos can be found. For example, table 2 in the text should be Table 2.

Please proofread the manuscript.

Answer: Thanks dear reviewer, we fully revised it again 

Figures: The format of figures (i.e., bar graphs) should be formatted in the same manner. One figure is 3D bar graph, and other is 2D bar graph.

Answer: Done dear reviewer, thanks.

Table 2: The horizontal lines on the top of the table is not normal. Thickness of horizontal lines should be corrected.

Table 3 : Thickness of horizontal lines should be corrected.

Answer: done dear reviewer, thanks.

Reviewer 2 Report

The authors have updated all the concerns and incorporated them in the appropriately revised manuscript. I accept the manuscript in this revised version.

Author Response

Open Review

English language and style

( ) English very difficult to understand/incomprehensible
( ) Extensive editing of English language and style required
( ) Moderate English changes required
(x) English language and style are fine/minor spell check required
( ) I don't feel qualified to judge about the English language and style

Yes

Can be improved

Must be improved

Not applicable

Does the introduction provide sufficient background and include all relevant references?

(x)

( )

( )

( )

Are all the cited references relevant to the research?

(x)

( )

( )

( )

Is the research design appropriate?

(x)

( )

( )

( )

Are the methods adequately described?

(x)

( )

( )

( )

Are the results clearly presented?

(x)

( )

( )

( )

Are the conclusions supported by the results?

(x)

( )

( )

( )

Comments and Suggestions for Authors

The authors have updated all the concerns and incorporated them in the appropriately revised manuscript. I accept the manuscript in this revised version.

Answer: Thanks once more for all your comments and suggestions dear reviewer.